# Viscosity jump in the lower mantle inferred from melting curves of ferropericlase

Jie Deng [1] & Kanani K.M. Lee [1]

Convection provides the mechanism behind plate tectonics, which allows oceanic lithosphere to be subducted into the mantle as "slabs" and new rock to be generated by volcanism. Stagnation of subducting slabs and deflection of rising plumes in Earth's shallow lower mantle have been suggested to result from a viscosity increase at those depths. However, the mechanism for this increase remains elusive. Here, we examine the melting behavior in the MgO–FeO binary system at high pressures using the laser-heated diamond-anvil cell and show that the liquidus and solidus of $(Mg_xFe_{1-x})O$ ferropericlase ($x = {\sim}0.52$–0.98), exhibit a local maximum at ~40 GPa, likely caused by the spin transition of iron. We calculate the relative viscosity profiles of ferropericlase using homologous temperature scaling and find that viscosity increases 10–100 times from ~750 km to ~1000–1250 km, with a smaller decrease at deeper depths, pointing to a single mechanism for slab stagnation and plume deflection.

[1] Department of Geology and Geophysics, Yale University, New Haven, CT 06511, USA. Correspondence and requests for materials should be addressed to J.D. (email: jie.deng@yale.edu)

Recently, Rudolph et al.[1] observed a jump in viscosity by a factor of 10–100 at 800–1200 km depths based on a reanalysis of the long-wavelength nonhydrostatic geoid, indicating a possible correlation between this viscosity variation and slab stagnation[2] and plume deflection[3,4] in the shallow lower mantle. Assuming an initially silica-rich lower mantle, a recent geodynamic modeling study[5] argued that a 20-fold change in viscosity associated with large-scale heterogeneity is sufficient to prevent efficient mantle mixing and generate the intrinsically strong bridgmanite-enriched domains in the shallow lower mantle, which in turn can explain the present day radial viscosity jump at those depths. However, the prescribed relationship between silica concentration and bulk viscosity of materials is rough as viscosity is not only sensitive to the major element chemistry but also many other factors including deformation mechanism, strain, and grain size. Alternatively, a change in the redox state of bridgmanite at pressures of 40–70 GPa may alter the Fe partitioning between bridgmanite and ferropericlase and result in an iron-depleted bridgmanite zone in the shallow lower mantle[6]. However, the effects of Fe on the melting temperatures of bridgmanite are controversial[7,8] and consequently the viscosity increase induced by the high melting temperatures of iron-depleted bridgmanite inferred from the homologous temperature scaling remains poorly constrained. Thus, although seismic tomography, geoid inversion and geodynamic modeling provide robust evidence for the viscosity jump in the mid-mantle, the corresponding mechanism remains enigmatic.

The viscosity of the lower mantle is suggested to be strain dependent[9]. In regions where large shear strain occurs (e.g., near subducting slabs and defected plumes), the weakest phase, (Mg, Fe)O ferropericlase, would form an interconnected weak layer (IWL) and therefore is expected to dominate the viscosity. Using homologous temperature scaling, the rheology of the lower mantle can be further assessed. The rate of plastic deformation of ferropericlase is proportional to the effective self-diffusion coefficients of the slowest species, i.e., Mg and Fe[9], which, in turn, are related to the melting temperature by the homologous temperature scaling relation. Using the melting curve of MgO reported by Zerr and Boehler[10], Yamazaki and Karato[9] derived a nearly depth-independent viscosity of ferropericlase. But recent experiments[11–13], first-principles calculations[14] and thermodynamic modeling[15] consistently favor a much higher melting curve of MgO with a much larger zero pressure melting slope ($dT_m/dP$, where $T_m$ is the melting temperature under pressure $P$). Additionally, because ferropericlase in the lower mantle likely contains 15–20% iron depending on the iron partitioning between bridgmanite[16], a linear reduction of melting temperature based on the percentage of iron is often carried out when calculating the melting curve of iron-bearing ferropericlase[17] without any physical basis. Therefore, better-constrained melting curves of iron-bearing ferropericlase are important to understand the variation of viscosity of ferropericlase at high pressures and temperatures.

Here, using the laser-heated diamond-anvil cell (LHDAC), we study the melting phase relations of the MgO–FeO binary system up to ~80 GPa. We use both ideal and regular solution models fit our experimental data and both suggest that the liquidus and solidus curves of $(Mg_xFe_{1-x})O$ ferropericlase ($x = \sim 0.52–0.98$), exhibit a local maximum at ~40 GPa. Based on these melting curves, the relative viscosity profiles of ferropericlase of Earth-relevant compositions are calculated using homologous temperature scaling. We find that the viscosity of ferropericlase shows a 10–100 times increase from ~750 km to ~1000–1250 km, and a subsequent smaller scale decrease at deeper depths, irrespective of deformation mechanism or mantle heterogeneity.

## Results

**Laser heating and chemical characterization.** We performed high-pressure melting experiments using the LHDAC on fine-grained $(Mg_xFe_{1-x})O$ ferropericlase ($x = 0.20, 0.23, 0.81, 0.82, 0.88, 0.90, 0.91$) at pressures up to 80 GPa (see "Methods" for details). Temperatures were determined using the inverse modeling method (see "Methods" for details). Samples were recovered from the LHDAC and examined with an analytical scanning electron microscope (ASEM) using wavelength dispersive spectroscopy (WDS). Sample cross-sections, chemical characterization of the run products, and determined temperatures can be found in Fig. 1, Supplementary Fig. 1, Supplementary Table 1, and Supplementary Note 1.

**Phase diagram calculation.** The mixing of MgO and FeO at low pressures (e.g., 3–7 GPa) has been described by the ideal solution model[11] for both liquid and solid states, whereas Frost[18] resolved non-zero interaction parameters (Margules parameters), $W_{FeO-MgO}^{solid}$ for MgO–FeO solid solution. Therefore, we used both the ideal solid solution and symmetric regular solution models to fit our data to avoid any inherent biases in the models. The biases are, for example, that the ideal solution model is incapable of producing inflection points in phase loops.

The equations we used to fit our data at each pressure are

$$\Delta H_{m,FeO}\left(1 - \frac{T}{T_{m,FeO}}\right) + W_{FeO-MgO}^{liquid}\left(X_{MgO}^{liquid}\right)^2$$
$$- W_{FeO-MgO}^{solid}\left(X_{MgO}^{solid}\right)^2 + RT\ln\frac{1 - X_{MgO}^{liquid}}{1 - X_{MgO}^{solid}} = 0, \quad (1)$$

$$\Delta H_{m,MgO}\left(1 - \frac{T}{T_{m,MgO}}\right) + W_{FeO-MgO}^{liquid}\left(1 - X_{MgO}^{liquid}\right)^2$$
$$- W_{FeO-MgO}^{solid}\left(1 - X_{MgO}^{solid}\right)^2 + RT\ln\frac{X_{MgO}^{liquid}}{X_{MgO}^{solid}} = 0, \quad (2)$$

where $R$ is the gas constant, $T$ is temperature, $\Delta H_{m,FeO}$ and $\Delta H_{m,MgO}$ are the enthalpy of melting of pure FeO and MgO, respectively. $T_{m,FeO}$ and $T_{m,MgO}$ are the melting temperatures of pure FeO and MgO and are directly taken from refs. [14,19] respectively. $X_i^{liquid/solid}$ is the component i (FeO or MgO) in the liquid or solid phase. $W_{FeO-MgO}^{solid}$ and $W_{FeO-MgO}^{liquid}$ the Margules parameters for the solid and liquid MgO–FeO solution, respectively. For the ideal solution models, the Margules parameters are assumed to be zero and only two paraeters, $\Delta H_{m,FeO}$ and $\Delta H_{m,MgO}$ are free fitting parameters. The regular solution models, in contrast, have four undetermined parameters. As the data set available for each pressure is limited, direct fitting for four parameters leads to large non-uniqueness. Therefore, extra constraints on the fitting parameters are necessary. $W_{FeO-MgO}^{solid}$ is informed by previous studies[20,21]. Frost et al.[20] gives the following equation:

$$W_{FeO-MgO}^{solid}(kJ\,mol^{-1}) = 11 + 0.11P, \quad (3)$$

where $P$ is pressure in GPa (the uncertainty is not given in the source). However, we note that this relation is based on only one experimentally determined $W_{FeO-MgO}^{solid}$ value at 18 GPa[18]. As such, we combine this value and another available experimentally determined $W_{FeO-MgO}^{solid}$ value for olivine[21] at 1 bar and fit them together with respect to pressure to get the following relation:

$$W_{FeO-MgO}^{solid}(kJ\,mol^{-1}) = 2.60(\pm 0.50) + 0.59(\pm 0.03)P, \quad (4)$$

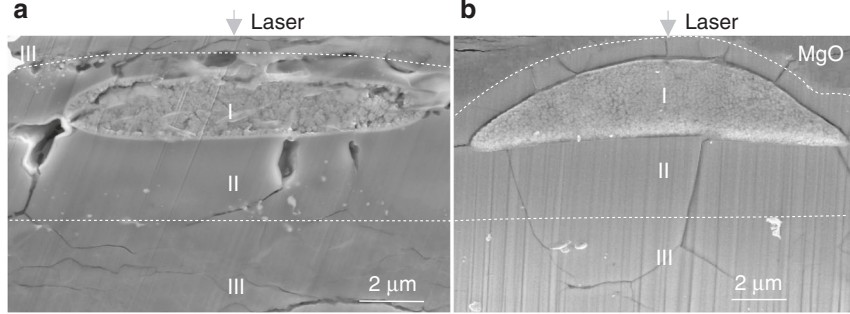

**Fig. 1** SEM images of representative cross sections of recovered ferropericlase melted samples. Samples melted at **a** 60 GPa (Jul0415) and **b** 27 GPa (Fp2002). Both images show a clearly quenched melt (I), co-existing solid that is relatively iron depleted (II), and starting material (III) regions. The starting material used in the left panel [$(Mg_{0.81}, Fe_{0.19})O$] was self-insulated whereas the starting material used in the right panel [$(Mg_{0.20}, Fe_{0.80})O$] was insulated by pure MgO. $(Mg_{0.20}, Fe_{0.80})O$ reacted with MgO, generating a more Mg-rich melt and coexisting solid compared with the starting material

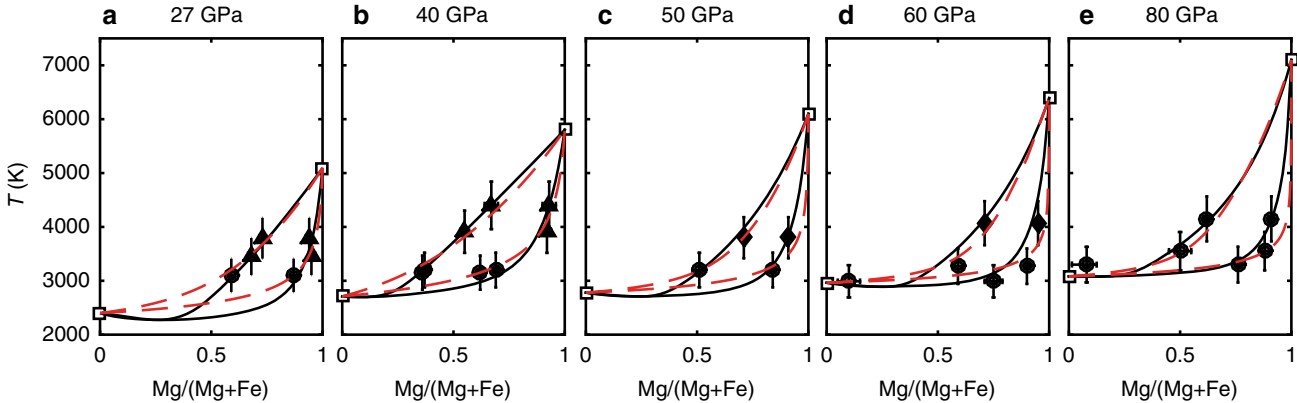

**Fig. 2** The phase diagrams of ferropericlase. Curves in **a**–**e** show the ideal (red, dashed) and regular solid solution models (black, solid) at 27, 40, 50, 60, and 80 GPa, respectively. The solid symbols represent the composition of melt and coexisting ferropericlase (solid circles, triangles and diamonds represent data from this study, refs. [12,49] respectively). Uncertainties (95% confidence interval) for temperatures and compositions are shown and listed in the Supplementary Table 1. Temperatures are measured and corrected using the multi-layer inverse modeling method (Supplementary Table 1)[49]. FeO and MgO melting temperatures (open squares) are taken from refs. [14,19] respectively

where $P$ is pressure in GPa. We use Eqs. (3) and (4) as the bounds for $W_{FeO-MgO}^{solid}$ values used in the fitting of the phase diagrams. The other Margules parameter $W_{FeO-MgO}^{liquid}$ has not been studied before in MgO–FeO system at high pressures and thus it is treated as a free fitting parameter.

As for the remaining two parameters, $\Delta H_{m,FeO}$ and $\Delta H_{m,MgO}$, we emphasize that they are the enthalpy change of endmembers at reference states. For example, $\Delta H_{m,FeO}$ is the enthalpy difference between the liquid and solid FeO at the melting temperature (here the reference temperature is the melting temperature of FeO). According to previous studies[22,23], FeO remains in the B1 structure along its melting temperatures to pressures well above our experimental range. Therefore, the structural phase transitions of FeO at low temperatures[24] are not relevant in fitting the solidus and liquidus loops. There are no direct experimental studies of $\Delta H_{m,FeO}$ at high pressures. Previous thermodynamic modelings[20,25] yield the equation of states of liquid FeO, which combined with the experimentally determined equation of states of solid FeO[23] enable us to calculate the $\Delta H_{m,FeO}$ at elevated pressures. The resolved $\Delta H_{m,FeO}$ values based on refs. [20,25] are almost identical at pressure lower than 11 GPa but become gradually discrepant with pressure and are different by ~20 kJ mol$^{-1}$ at 80 GPa. We adopt both $\Delta H_{m,FeO}$ values as the bounds in phase diagram modeling using the regular solution model. In contrast, both $\Delta H_{m,MgO}$ values and their trends with respect to pressure reported by first-principles

calculation[14,26,27] at pressures greater than 1 bar are extremely discrepant and the discrepancies increase with pressure. For example, at 90 GPa the smallest $\Delta H_{m,MgO}$ value reported by the calculation study is only ~75 kJ mol$^{-1}$ [20] in contrast to the largest value of ~180 kJ mol$^{-1}$ [14]. Therefore, we set $\Delta H_{m,MgO}$ as a free fitting parameter.

The best fitting parameters are shown in Supplementary Tables 2 and 3 for the ideal and regular solution model, respectively. Although the ideal and regular solution models yield similar phase diagrams, the ideal solution model gives unrealistically high $\Delta H_{m,FeO}$. This is because $\Delta H_{m,FeO}$ is related $\Delta V_{m,FeO}$, the volume change of melting by $\Delta H_{m,FeO} = \Delta V_{m,FeO} \times T_m \frac{dP}{dT_m}$. The $\Delta H_{m,FeO}$ at 1 bar has been experimentally determined to be ~35 kJ mol$^{-1}$ [28], corresponding to $\Delta V_{m,FeO}$ of ~0.52 cm$^3$ mol$^{-1}$. A $\Delta H_{m,FeO}$ of 105 kJ mol$^{-1}$ at 27 GPa indicates that $\Delta V_{m,FeO}$ is ~1.2 cm$^3$ mol$^{-1}$, which is unlikely considering that the compression of both solid and liquid FeO at high pressures. Therefore, a regular solution model might be more appropriate to describe the MgO–FeO solution in the pressure range we examined.

The phase diagrams at 27, 40, 50, 60, and 80 GPa constructed using the best fitting parameters are shown in Fig. 2. The ideal solution model and regular solution model yield overall similar phase diagrams, especially at higher pressures. Additionally, we also apply our parameters fitted at high pressures but interpolated to low pressures, to existing literature data at 3 GPa[12]

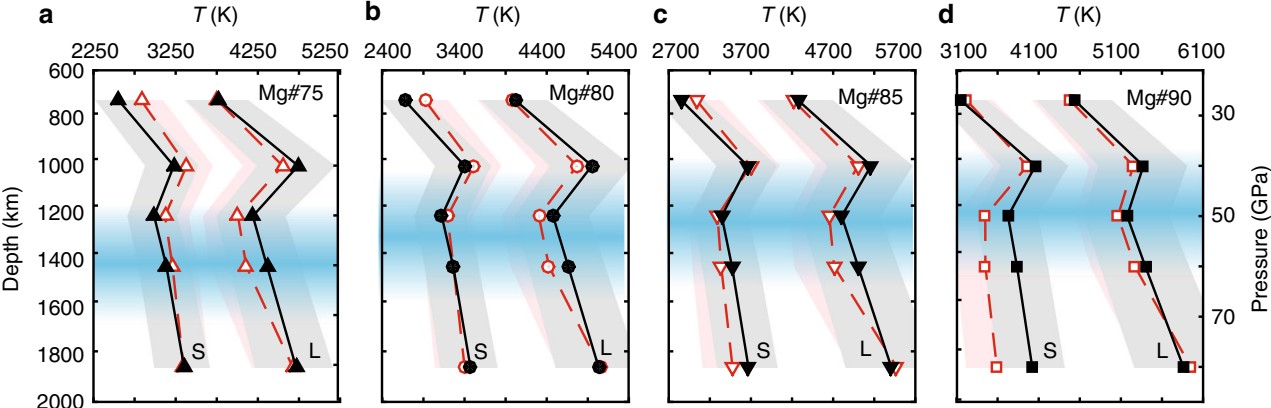

**Fig. 3** Melting curves of ferropericlase. Liquidus (L) and solidus (S) temperatures of **a** $(Mg_{0.75}Fe_{0.25})O$ (triangles), **b** $(Mg_{0.8}Fe_{0.2})O$ (circles), **c** $(Mg_{0.85}Fe_{0.15})O$ (inverted triangles), and **d** $(Mg_{0.9}Fe_{0.1})O$ (squares) assuming the ideal (red, open symbols) and regular solution models (black, solid symbols), respectively. Uncertainties (95% confidence interval) on the temperatures are ~±10% as indicated by the red and gray shaded regions for the ideal and regular solution models, respectively. The blue shaded regions correspond to the composition-dependent spin transition pressure range at 300 K (~35–70 GPa)[33]. The spin transition pressure ranges at corresponding high temperatures are likely broader but begin at pressures similar to those at 300 K[65,66]. The curves are drawn as a guide for the eye

and find good agreement (Supplementary Fig. 2 and Supplementary Note 2), helping to validate our solution models.

**Melting curves of ferroepericlase.** Based on the phase diagrams, we infer the solidus and liquidus curves for (Mg,Fe)O with different Earth-relevant iron contents (Fig. 3). Four consistent features of the solidus and liquidus curves of ferropericlase are noticeable. First, the solidus temperatures of ferropericlase are much smaller (more than 1000 K) than those extrapolated by a linear reduction of melting curves of pure MgO and FeO[14] (Supplementary Fig. 3 and Supplementary Note 3). Second, the melting temperatures (both solidus and liquidus) locally peak at ~40 GPa and the difference between the melting temperatures at 27 and 40 GPa is as high as ~900 K. Third, the magnitude of the melting temperature (both solidus and liquidus) increase from 27 to 40 GPa (i.e., melting slope) is larger than that of the melting temperature decrease from 40 to 50 GPa. Fourth, between 50 and 60 GPa, the melting slope becomes positive again. It should be noted that most of our experimental data set is concentrated in the Mg-rich end. Therefore, the ferropericlase melting temperatures with Earth-relevant compositions (Mg#=100×Mg/(Mg+Fe) by mol=75–90) are robustly constrained by the experimental data regardless of the solution models implemented (Fig. 2).

## Discussion

Our preferred way to interpret the observed unusual melting behavior of (Mg,Fe)O is the spin crossover of $Fe^{2+}$ in (Mg,Fe)O. The spin transition of iron in the lower mantle is generally expected to result in considerable changes including: physical (e.g., density and elastic moduli), chemical (e.g., element partitioning), and transport properties (e.g., electrical conductivity)[29]. While it is unknown how the spin transition affects the melting of (Mg,Fe)O, a qualitative estimate can be obtained based on Lindemann's law, which provides a simple relationship between the melting temperature and the thermoelastic properties of materials, $T_m \propto C/\rho$, where $C$ is some combination of elastic moduli and $\rho$ is the density[9,30]. Both experiments and first principles computations have shown that the spin transition softens and densifies (Mg,Fe)O[31,32]. Therefore, we can expect that the spin transition will tend to lower the melting temperatures based on Lindemann's law (see Supplementary Fig. 4 and Supplementary Note 4 for further discussion), setting up a local

maximum in each melting curve. However, after the mid-point of the spin crossover, the moduli will again monotonically increase[31,32], thus causing the melting temperatures to again increase. For Earth-relevant compositions, the local maximum in $T_m$ occurs at ~40 GPa based on the spin-state crossover region[33]. Additionally, the varying signs of melting slopes for the melting curves of ferropericlase is unexpected but not unique[34] and is consistent with the spin crossover of $Fe^{2+}$ in both (Mg,Fe)O solid[35] and liquid[36,37] (Supplementary Note 5).

With the melting curves of ferropericlase, we further calculated the effective diffusion coefficients using homologous temperature scaling. As has been argued for multi-component systems, we use the solidus as the melting temperature in the homologous temperature scaling[30]. The radial relative viscosity profiles of ferropericlase based on its solidus curves are shown in Fig. 4 using both a "cool" geotherm[38] and a relatively "hot" geotherm[39]. We also calculate the radial relative viscosity profiles of ferropericlase based on the liquidus curves and find similar results (Supplementary Fig. 5).

Our results are in contrast to early high-pressure, room-temperature deformation experiments[40,41] on $(Mg_{0.83}Fe_{0.17})O$ and MgO which show negligible rheological variations with pressure. More recently, a deformation experiment on $(Mg_{0.9}Fe_{0.1})O$ and $(Mg_{0.8}Fe_{0.2})O$ at elevated temperatures (up to 770 K), found that the strength of ferropericlase increases by a factor of three at pressures from 20 to 65 GPa[17]. Further modeling based on this strength increase suggests a viscosity jump by ~2.3 orders of magnitude in regions of large shear strain in the shallow lower mantle. While their result[17] is roughly consistent with our results, at least until ~40–50 GPa, it is dependent on deformation mechanism and the data are collected at low temperatures, which makes the results difficult to apply to Earth's interior. Additionally, we also suggest the viscosity of ferropericlase should decrease between ~40 and 50 GPa, while Marquardt and Miyagi[17] indicate that the viscosity of ferropericlase will continue to increase, likely due to the low temperatures achieved in their experiments.

It is noted that the viscosity profile of ferropericlase calculated in this study is in excellent agreement with that proposed by Wentzcovitch et al.[42], where viscosity was estimated based on the elastic strain energy model. The viscosity is a function of the effective diffusivity (see Eqs. (8) and (9) in "Methods") which is further related to the shear and bulk moduli by the elastic strain

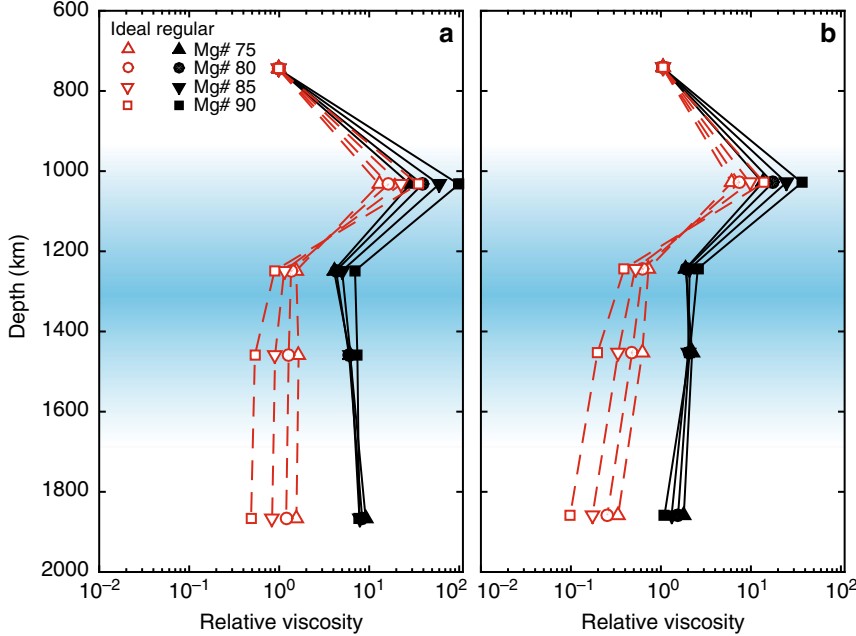

**Fig. 4** The radial relative viscosity profiles of ferropericlase inferred from homologous temperature scaling and the solidus melting curves. Profiles based on the ideal (red, open symbols) and regular solution models (black, solid symbols) for $(Mg_{0.75}Fe_{0.25})O$, $(Mg_{0.8}Fe_{0.2})O$, $(Mg_{0.85}Fe_{0.15})O$ and $(Mg_{0.9}Fe_{0.1})O$ assuming a **a** "cool" geotherm[38] and a relatively **b** "hot" geotherm[39]. Symbol identification is the same as in Fig. 3. Viscosity values are scaled relative to the viscosity at 27 GPa. For these curves, the non-dimensional constant $g$ in the homologous temperature scaling is given a value of 12 (Methods). Uncertainties on the viscosity are approximately enveloped by the values given by the solidus and liquidus (Supplementary Fig. 5) and two different solid solution models under two different geotherms. The blue shaded regions include spin transition pressure range for Mg#75-90 at 300 K (~35–70 GPa)[33]. The spin transition pressure ranges at corresponding high temperatures are likely broader but begin at pressures similar to those at 300 K[65,66]. The curves are drawn as a guide for the eye

energy model because ionic diffusion shears and stretches the chemical bonds on the diffusion path[42]. Using the elastic constants obtained by first-principle computations, a local peak viscosity was found at ~40 GPa[42]. The consistency between ref. [42] and this study is not fortuitous. Both studies suggest that the viscosity of ferropericlase is correlated with its thermoelastic properties. Although our observation is only directly concerned with the melting behavior of ferropericlase, melting has been suggested to be closely related to the thermoelastic properties in general by both the shear instability model and the vibrational instability model (see ref. [43] for a review). Based on the latter instability model, Lindemann's law mentioned earlier is one example that relates the elastic constants to melting temperatures. As such, the viscosity profile of ferropericlase by ref. [42] and this study indicate that the viscosity peaking at ~40 GPa is a necessary result of the thermoelastic anomaly induced by the spin crossover of $Fe^{2+}$ for Earth-relevant ferropericlase.

Note that in our analysis, diffusion creep and dislocation creep yield identical relative viscosity profiles. This is because we do not include the strength variation suggested by the low temperature deformation experiments for dislocation creep[17] and consequently for both dislocation creep and diffusion creep, the resultant viscosity of ferropericlase is $\eta \propto T/D_{eff}$, where $D_{eff}$ is the effective diffusion coefficient of Mg or Fe and $T$ is temperature. While the specific values of the viscosity of ferropericlase depends on the values we use for the input parameters in the homologous temperature scaling and in the formula for different deformation mechanisms ("Methods"), a viscosity jump by one to two orders of magnitude from 27 GPa (~750 km) to ~40–50 GPa (~1000–1250 km) is a necessary result of the corresponding increase in melting temperature of ferropericlase with the compositions examined here regardless of the dominant

deformation mechanisms and geotherms used. Likewise, a viscosity drop of smaller scale at deeper depths is also a necessary result stemming from the ferropericlase melting curves.

In large strain regions where ferropericlase likely forms an IWL framework[44], a one to two orders of magnitude jump in viscosity of ferropericlase results in nearly the same scale of jump in the viscosity of the lower mantle. This offers a simple mechanism for the sharp viscosity increase in the shallow lower mantle and the corresponding depth of the maximum viscosity is in excellent agreement with the maximum depth at which most actively subducting slabs stagnate[2]. Since the viscosity jump predicted by this study does not rely on the deformation mechanism of ferropericlase, if we include the contribution of an elastically strengthened ferropericlase to the viscosity[17] assuming the deformation mechanism is dislocation creep as suggested by ref. [45], the viscosity further increases to two to four orders of magnitude, up from one to two.

For the majority of the shallow lower mantle away from stagnant slabs and deflected plumes where the shear strain is likely relatively small and seismic anisotropy is generally not observed[46], our analysis suggests that the viscosity of ferropericlase still increases by at least one to two orders of magnitude. But whether this jump in viscosity for ferropericlase results in the bulk mantle viscosity increase by a similar scale strongly depends on the strain partitioning between bridgmanite and ferropericlase. The IWL framework is an extreme case in which most of the strain preferentially partitions into the weak phase, and thus the weak phase dominates the deformation rate of the aggregate. In contrast, strain equally partitions between the strong and weak phases in a load-bearing framework (LBF) where the strong phase containing isolated pockets of the weak phase. The degree of strain partitioning into a weaker phase has been suggested to increase with the strain under plastic deformation[47].

If the shear strain in the majority of the shallow lower mantle is relatively small so that LBF dominates, the viscosity is dominated by the dominant and more viscous phase (bridgmanite) and the viscosity jump of ferropericlase in this region will be overshadowed.

Our conclusion is comparable with Marquardt and Miyagi[17] in that both studies predict a sudden jump of the viscosity of ferropericlase in the shallow lower mantle. As such, each can explain the broadening and stagnation of subducting slabs at those depths. The difference between our results and the previous study's[17] is at pressures greater than 40 GPa and is likely due to the fact that their experiment is conducted at relatively low temperatures and is not directly applicable to the high-temperature rheology of ferropericlase in Earth's deep interior. Additionally, our results yield a natural explanation for plume deflection at depths of ~1000 km[3,4] because of viscous resistance for upwelling plumes also peaks at this depth. These conclusions do not hinge on the assumption that the dominant deformation mechanism in large strain areas in the shallow lower mantle is dislocation creep. Additionally, our results show the existence of this viscosity peak is relatively insensitive to the major element chemistry of ferropericlase for plausible lower mantle ferropericlase compositions. Therefore, the intrinsic viscosity profile inferred from the melting curve of ferropericlase likely affects the pattern of mantle convection in the Earth.

## Methods

**Sample synthesis and laser heating experiments**. $(Mg_{1-x}Fe_x)O$ ferropericlase ($x = 0.09, 0.2, 0.23, 0.82, 0.88, 0.90, 0.91$) were synthesized from powders of $Fe_2O_3$ and fired MgO which were annealed for 14 h at 1473 K at an oxygen fugacity of $10^{-5}$ Pa[12,48]. In order to minimize chemical reactions between widely used pressure media and ferropericlase, Mg-rich ferropericlase starting materials were loaded without any pressure transmitting medium into LHDACs equipped with either matched 150, 200, or 300 µm culets. For example, using a noble gas such as argon may cause lowered melting temperatures due to incorporation in to melts of this composition[12]. Additionally, due to the high temperatures anticipated for ferropericlase melting, we avoided alkali halides because of strong changes in the optical properties[49], which cause a rapid increase in temperature near their melting points[50]. See Supplementary Note 6 for a thorough discussion about the absence of the pressure medium for some experiments in this study. For Fe-rich ferropericlase samples, ferropericlase must be insulated in order to protect the diamond anvil on the heated side. We therefore loaded a layer of pure MgO powder between the diamond anvil and the Fe-rich ferropericlase on the heated side, which also helped to minimize axial temperature gradients[49] and contamination from other potential pressure media. In order to minimize moisture contamination, the starting materials were kept in a vacuum oven (80 °C) except when taken out for sample loading. Additionally, the sample assembly was also oven-dried after cell loading and prior to pressurization. Pressurized samples were heated from one side with a near-infrared laser (100 W, 1070 nm SPI water-cooled fiber laser). We used pre-defined ramped laser heating[12] to melt the sample: the laser is set to a low power for 2 s and then linearly ramped up to a peak power every 20 ms within 1 s and kept at the peak power for 0.4–1 s before turned off. A mechanical shutter is opened ~40–100 ms before the laser is quenched to allow the temperature measurement of the sample at peak power.

**Chemical analyses**. Quenched samples were cut and polished through the center of the heated region by electrical discharge machining and focused ion beam (FIB) techniques (Fig. 1 for representative cross-section images), and then quantitatively analyzed by an ASEM (JEOL 7600F) using WDS. Both pure iron (purity > 99.95%) and (Mg,Fe)O samples with known compositions were selected as reference standards for WDS measurements. Chemical analyses were performed using an accelerating voltage of 10 KeV and a beam current of 6.5 nA, with spatial resolution of ~1 µm in diameter[51].

The measurement totals of our quenched samples are mostly within the range of 97–103%, suggesting that contamination from carbon, if it exists, is likely small. In addition, previous studies suggest a negligible solubility of carbon in (Mg,Fe)O. Shcheka et al.[52] found that the maximum carbon solubility in most mantle silicates, including wadsleyite, ringwoodite, $MgSiO_3$—ilmenite and $MgSiO_3$—bridgmanite is below their limit of detection of 40–110 ppb by weight. Because (Mg,Fe)O does not have tetravalent cation, we expect that the solubility of carbon in ferropericlase should be lower than the above silicates and therefore negligible. Indeed, the solubility of carbon in FeO and MgO solid was reported under the detection limit (0.01 ppm by weight) at 1 bar[53]. Although the effects of carbon on the melting of

ferropericlase is unknown, a carbon concentration <0.01 ppm is unlikely to have any pronounced effects on the melting of (Mg,Fe)O ferropericlase.

**Temperature determination**. Accurate temperature measurements using spectroradiometric approaches[54] or multi-wavelength two-dimensional imaging radiometry[55,56] for semi-transparent materials in LHDAC experiments is difficult, because wavelength-dependent absorption/emission and temperature gradients at elevated pressures may drastically deviate the apparent temperature from the real highest temperature attained during the experiments[49]. The optical absorption spectra of (Mg,Fe)O ferropericlase exhibit strong wavelength dependency which varies significantly with Fe content, pressure and temperature[57]. Consequently, the wavelength-dependent absorption must be taken into account when determining temperatures of (Mg,Fe)O samples in LHDAC experiments. The wavelength-dependent absorption issue cannot be simply solved by using thin samples within a thick and inert insulating medium. On the contrary, a thin sample will make the situation worse because the temperature deviation can easily be very large when the optical thickness of the sample $\tau_\lambda = \int_0^d k_\lambda dz$ is small, where $k_\lambda$ is the absorption coefficient and $d$ is the thickness of materials that participate in the radiative heat transfer at the wavelengths of interest. A thinner sample is characterized by a smaller optical thickness and therefore a large temperature deviation is more likely (see Fig. 4 in ref. [49]). Here, we implemented the multi-layer inverse modeling method in which the geometry and optical properties of the sample and detected thermal radiation intensities during each experiment are integrated to rigorously constrain the melting temperatures of ferropericlase[49]. The results (inverse modeling $T_m$) are tabulated in Supplementary Table 1. Apparent temperatures ($T_a$) calculated by averaging the temperatures along the loop encompassing the molten region (region I in the Supplementary Fig. 1a) in 2D optical images (perimeter $T_a$) and apparent temperatures of position with strongest thermal radiation intensities (hottest point $T_a$) are also shown for comparison.

At first glance, the differences between $T_m$ and $T_a$ do not exhibit any obvious systematic trends with respect to pressure or composition (Supplementary Table 1). However, there is a qualitative prediction of the trend. For example, samples Jul0415 ($T_m - T_a = 670$ K) and 14_0506_45G ($T_m - T_a = -660$ K) have opposite temperature corrections. It is worth noting that in the wavelength range of thermal radiation we measured (580, 640, 766, and 905 nm), (1) the optical absorption coefficients of (Mg,Fe)O ferropericlase generally decrease with wavelength, and (2) both the absorption coefficients and the slopes of the optical absorption spectra exponentially increase with Fe concentration of ferropericlase[57]. For sample Jul0415, the melt is very iron rich and relatively thick so that the optical thickness in the four wavelengths measured (580, 640, 766, and 905 nm) are very large and the melt can be treated as a blackbody. Therefore, the radiation emitted from the melt is nearly intact. But after the radiation penetrates through the overlaying coexisting solid and starting materials and is recorded, the short wavelength radiation signal (high energy) is preferentially absorbed, yielding a radiation spectrum with the intensities at short wavelengths undermined. Consequently, the later Wien or Planck fitting based on this biased spectrum gives a cooler apparent temperature. However, for sample 14_0506_45G, the melt is not as iron rich and relatively thin as compared to Jul0415, such that the optical thickness in 580, 640, 766, and 905 nm is relatively smaller. According to Kirchoff's law, the thermal radiation from the melt itself is biased in the way that high-energy signals (short wavelength) are relatively enhanced because of the large absorptivity at short wavelengths. Although the preferential absorption of short wavelength radiation by the overlying coexisting solid and the starting material can mitigate the effects to some extent, the coexisting solid and the starting material are too transparent (Mg#93 and 90, respectively) and thin to make a big difference. Consequently, the overall effect is dominated by the melt and the thermal radiation detected exhibits a hotter apparent temperature.

While the above qualitative analysis can explain the general trend of $T_m$–$T_a$, a rigorous evaluation of temperature requires careful inverse modeling taking into account the geometry, optical properties of the sample and temperature gradients, as shown in Supplementary Fig. 1. The best fitting horizontal intensity of radiation curves (thin lines in Supplementary Fig. 1d) are generally very consistent with the detected data (thick lines) except that the fitted results are overall slightly higher than the detected values, which might be a result of inaccurate interpolation or extrapolation of the absorption coefficients at a particular wavelength. Additionally, the fitted curves become increasingly disparate from the real data at the ends. This is reasonable since the white lines in Supplementary Fig. 1c do not delineate the layer boundaries at the two ends very well and are only an approximation to the actual shape.

**Viscosity calculation using homologous temperature scaling**. Homologous temperature scaling relates the effective diffusion coefficients at high pressures to melting temperatures by,

$$D(P, T) = D_0 \exp(-gT_m(P)/T), \qquad (5)$$

where the pre-exponential factor $D_0$ is a constant, $T_m(P)$ is the melting temperature at pressure $P$, and $g$ is a non-dimensional constant and is given by

$$g = H^*(P)/RT_m(P), \qquad (6)$$

where $H^*(P)$ is the activation enthalpy at a given pressure $P$. Following previous studies[9,17], we derive the $g$ values by taking the zero pressure $H^*(P)$ from computational studies[58–61] and zero pressure solidus melting temperature of $(Mg_{0.80}Fe_{0.20})O$ by phase diagram calculation using the regular solution model. The resultant $g$ values are 10–14. Taking the derivative of $H^*(P)$ with respect to pressure yields the activation volume,

$$V^* = gR(dT_m/dP). \qquad (7)$$

For Earth-relevant compositions (Mg#75, 80, 85, 90), our inferred average activation volumes at pressures 3–27 GPa and 60–80 GPa are consistent self-diffusion activation volumes of Mg in pure MgO[58,59,62] (Supplementary Fig. 6). As discussed above the spin transition of $Fe^{2+}$ in solid $(Mg,Fe)O$ of those compositions at 300 K starts and completes at ~35 and ~70 GPa, respectively[11,63] and the spin transition pressure range in solid $(Mg,Fe)O$ counterparts might be smaller[37]. The overall effects of spin transition in both liquid and solid $(Mg,Fe)O$ is that the $V^*$ at ~40 and 50 GPa is strongly affected as shown in Supplementary Fig. 6. Nevertheless, the general good agreement between the activation volumes of self-diffusion in $(Mg,Fe)O$ inferred based on homologous temperature scaling and those in MgO resolved by experiments and computations strongly validates the use of the homologous temperature scaling method in this study. As such, we use homologous temperature scaling to infer viscosity in the mantle as shown below.

Under dislocation creep, the viscosity of ferropericlase is expressed as[17,64]

$$\eta = A_{dis}\frac{RT}{D\tau_p\mathbf{b}}\left(\frac{\tau_p}{\sigma}\right)^n, \qquad (8)$$

where $A_{dis}$ is a pre-exponential factor, $D$ is the lattice diffusion coefficient of the slowest species, $\tau_p$ is Peierls stress, $\mathbf{b}$ is Burgers vector, $\sigma$ is stress, and $n$ is the stress exponent.

Under diffusion creep, the viscosity of ferropericlase is expressed as[9]

$$\eta = A_{dif}\frac{RT}{D\Omega}d^2, \qquad (9)$$

where $A_{dif}$ is a pre-exponential factor, $\Omega$ is the molar volume, and $d$ is the grain size of the constituent materials.

Here we only focus on the ratio of viscosities since the absolute value of viscosity is highly sensitive to the specific values of input parameters, e.g, $A_{dis}$, $A_{dif}$, $\tau_p$, $\sigma$, $n$, and $d$ of which most are poorly understood at lower mantle conditions. We therefore assume $A_{dis}$, $A_{dif}$, $\tau_p$, $\sigma$, $n$, and $d$ are constant from 27 to 80 GPa. The molar volume of ferropericlase, $\Omega$ slightly decreases with pressure (~10% reduction from 27 to 80 GPa for ferropericlase of the composition we examine here). We use the equation of state of $(Mg_{0.80}Fe_{0.20})O$ to calculate $\Omega$[36]. With above set-up, the viscosity ratio of ferropericlase at high pressure over the viscosity at reference state $P_0$ is

$$\frac{\eta(P)}{\eta(P_0)} = \frac{T(P)/D(P)}{T(P_0)/D(P_0)} = \frac{T(P)}{T(P_0)}\frac{\Omega(P_0)}{\Omega(P)}\exp\left(g\left(\frac{T_m(P)}{T(P)} - \frac{T_m(P_0)}{T(P_0)}\right)\right), \qquad (10)$$

where $T(P)$ is the geotherm.

Here we use a typical lower mantle geotherm[38] and set $P_0 = 27$ GPa. Taking the solidus as $T_m$, the radial relative viscosity profile of $(Mg_{0.80}Fe_{0.20})O$ is shown in Supplementary Fig. 5. The ratio of viscosity at ~1000 km over viscosity at ~750 km increases with g and is ~20 at $g = 10$ and is ~70 at $g = 14$. The relative viscosity profiles inferred from liquidus curves rather than solidus curves by both ideal and regular solution models are also shown in Supplementary Fig. 5, which in general, are in good agreement with those inferred from the solidus curves.

**Data availability**. The data sets generated during and/or analyzed during the current study are available as Supplementary Information and from the corresponding authors.

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

## Acknowledgements

We thank Z. Du, M. Long, S. Karato and J. Korenaga for helpful discussions; M. Rooks and F. Camino for FIB help at YINQE at Yale University and CFN at Brookhaven National lab, respectively; Z. Jiang for SEM assistance; J. Girard for laser heating assistance; Z. Liu for assistance in the collection of optical spectra of ferropericlase. This work was supported by NSF (EAR-1321956, EAR-1551348). FIB use was supported by YINQE and NSF MRSEC DMR 1119826 and by the Center for Functional Nanomaterials, and National Synchrotron Light Source II at Brookhaven National Laboratory, a U.S. Department of Energy (DOE) Office of Science User Facility operated for the DOE Office of Science under Contract No. DE-SC0012704. This research was also partially supported by COMPRES, the Consortium for Materials Properties Research in Earth Sciences under NSF Cooperative Agreement EAR 1606856.

## Author contributions

J.D. executed the experiments and analyses. J.D. and K.K.M.L. contributed equally to the design of the study and writing of the manuscript.

## Additional information

**Competing interests:** The authors declare no competing financial interests.

