## [Peer Review File · Nature Communications]

Reviewers' comments:

Reviewer #1 (Remarks to the Author):

This paper reports good and important measurements of the melting curve of ferropericlase in a large range of concentrations. It goes to model the homologous phase diagram.

1) It is understood the authors are interested in the melting behavior but they should clarify the nature of the solid phase for low Mg#. FeO undergoes a structural phase transition at or below ~ 20 GPa at 300 K. This invalidates the phase diagram modeling using a single solid phase in for large iron concentrations. The authors should justify their procedure further or restrict the modeling to large Mg#.

2) Despite this issue with the modeling, based on hard core data presented in Figure 2, it is pretty clear that at 40 GPa (it could be slightly different) the melting temperature (solidus or liquidus) is the largest for the relevant range of concentrations in the Earth. The authors then continue by relating this melting temperature with viscosity and predict a peak in viscosity at ~ 40 GPa, followed by a decrease up to ~ 50 GPa, followed again by a steady increase. If such a scaling relation between viscosity and melting temperature is valid, this is an exceedingly interesting behavior that contrasts with measurements of viscosity based deformation behavior at low temperatures (Ref. 16). This is a significant result.

3) The authors admit it is unclear how the spin crossover affects the viscosity. In a previously and very relevant publication (Fig. 4 in Ref. 26 in SI) another phenomenological estimate of viscosity based on the elastic strain energy model was used to speculate on the possible existence of a viscosity drop in the lower mantle above 40 GPa followed by an increase in the pressure range measured here. The cause of this behavior was discussed there as well and was related to the behavior of the activation energy for diffusion in the presence of a softening of the bulk modulus (or bond strength). It is very interesting that similar behavior for the viscosity is found now based on the melting behavior of ferropericlase, which is quite different from that put forward by Ref. 16. The similarities between the present results and that predicted in Ref. 26 of SI (Wentzcovitch et al, PNAS 2008) should be presented and discussed in the main text. These results support each other in contrast to the distinct viscosity behavior proposed in Ref. 16. This discussion is very important and should be included.

4) Details: Line 108: "...the spin transition softens and condenses (MgFe)O"... "Densifies" is a better expression than "condenses".

5) Details: Lines 111-112: "...once the spin transition to low-spin state is complete, the moduli and density will again monotonically increase at values greater than those in the high spin state...". Actually, this is incorrect. The moduli and density start increasing monotonically after the mid-point of the spin crossover, not after it has ended.

6) Details: Line 92: punctuation or parenthesis missing.

In summary, this paper presents good and important results on the melting behavior of ferropericlase. It is original and interesting data. As always, speculations of geophysical implications may not endure, as often is the case, but it illustrates the significance of these results to geophysics.

Reviewer #2 (Remarks to the Author):

General comments:

This article presents experimental measurements of the (Fe,Mg)O melting relations under high

pressure and a viscosity model for the Earth's mantle inferred from these results using homologous temperature scaling. This subject is suitable for Nature Communications. The presentation is well-written and clear. The experimental results are quite clear, even if the absence of pressure medium to determine melting properties needs more careful discussion. As a non-specialist regarding the model used here to relate melting curves and viscosity, it is difficult for me to judge the uncertainties for viscosity determination are. In conclusion, I would recommend this paper for publication with the addition of more complete discussion on the two previous points.

Detailed comments:

The experimental determination of the Mg,FeO phase diagram and melting relations under extreme conditions is a very challenging topic. I find quite clear the discussion regarding temperature determination, and I should compliment the authors for the past effort made on the wavelength dependent absorption of the sample.

I am however quite puzzled by the absence of pressure medium in the present melting experiments. I wonder how the temperature gradients generated could contribute somehow to the error bars and then affect the presented maximum in the melting curves. Also, is there any quantified C contamination in the sample?

Also, the absence of any irregular effect on FeO and MgO melting curves in the present pressure range is difficult to correlate, in my opinion, with the present maximum in MgFeO melting, even for low Mg content.

Reviewer #3 (Remarks to the Author):

Deng and Li, 2017 Nature Communications

reject

The authors carried out high-pressure melting experiments on (Mg,Fe)O ferropericlase (and magnesiowüstite) up to 83 GPa and constrained melting phase relations of MgO-FeO system. The obtained solidus and liquidus curves exhibit local maxima at about 40 GPa, which may cause viscosity jump in the lower mantle.

Since (Mg,Fe)O is second most abundant mineral in the Earth's lower mantle, its melting temperature could affect the melting temperature and viscosity of the lower mantle. I agree that the wavelength-dependent absorption must be considered in order to determine temperature by spectroradiometric method. Thermodynamic treatment to get phase relation of (Mg,Fe)O seems valid. However, since viscosity is not only sensitive to the melting temperature of consisting minerals, but also many other factors as the authors stated in the manuscript, the local maxima in melting curve of ferropericlase has been just a possible cause of the jump in viscosity. The present data is new, but I find no new scientific ideas in this manuscript. This manuscript should be submitted to professional journals such EPSL, PEPI and JGR rather than Nature Communications.

In addition, I have some minor comments on this manuscript.

1. Fine-grained (Mg,Fe)O samples were used in this study. Did the authors take care with contamination of moisture in the samples? Water in the sample drastically decreases its melting temperature as we know. Also, I wonder if the authors checked carbon contamination during laser heating experiments by using in-situ XRD or ex-situ chemical analyses.
2. Interpretation of the observed unusual melting behavior at ~40 GPa may be the spin crossover in (Mg,Fe)O. Spin transition zone spreads with increasing temperature as many experimental and theoretical studies reported (e.g., Lin et al., 2007 Science; Tsuchiya et al., 2004 PRL). However, the authors referred to the spin transition pressure range of 300 K as shown in Figs. 2 and 3. The authors discuss pressure response of melting curves, and should refer the spin transition zone at

high temperatures.

3. As discussed in Ohta et al. (2017 EPL 465, 29-37), within such a spin transition zone of ferropericlase, ferropericlase can be regarded as three component mixtures of MgO, high-spin FeO, and low-spin FeO, which induce the reduction of melting temperature of ferropericlase.

4. Holmstrom and Stixrude (2015 PRL) suggested metallization of ferropericlase with 25 mol% Fe at high pressures and temperatures. Is the metallization of ferropericlase another cause of anomaly in melting curves that the authors observe?

5. There is one new paper regarding experimentally-determined melting curve of MgO. Kimura et al., 2017 Nature Communications, Melting temperatures of MgO under high pressure by micro-texture analysis. I ask the authors to cite it.

6. Error bars of temperature should be added in Figs. 2 and 3. In Figs. 2 and 3, spin transition range of high temperature should be shown rather than that at 300 K.

Response

Reviewer #1 (Remarks to the Author):

This paper reports good and important measurements of the melting curve of ferropericlase in a large range of concentrations. It goes to model the homologous phase diagram.

1) It is understood the authors are interested in the melting behavior but they should clarify the nature of the solid phase for low Mg#. FeO undergoes a structural phase transition at or below ~ 20 GPa at 300 K. This invalidates the phase diagram modeling using a single solid phase in for large iron concentrations. The authors should justify their procedure further or restrict the modeling to large Mg#.

Thank you for the suggestions. Indeed, FeO undergoes multiple phase transitions with increasing pressures and temperatures [Fischer *et al.*, 2011; Ohta *et al.*, 2014] whereas MgO solid remains in the B1 structure at pressures up to ~600 GPa [Coppari *et al.*, 2013]. However, we emphasize that what is involved in our phase diagram modeling is the thermodynamic properties of endmembers at reference states. For example, $\Delta H_{m,FeO}$ is the enthalpy difference between the liquid and solid FeO at melting temperature (here the reference temperature is the melting temperature of FeO). According to [Ohta *et al.*, 2014], FeO remains in the B1 structure along its melting temperatures up to ~240 GPa. Therefore, the structural phase transition of FeO at ~20 GPa and 300 K is not relevant in fitting the solidus and liquidus loops. Additionally, as suggested by the reviewer below and mentioned in the manuscript, most of our experimental dataset is concentrated in the Mg-rich end. Therefore, the ferropericlase melting temperatures with Earth-relevant compositions (Mg# = $100 \times \text{Mg}/(\text{Mg} + \text{Fe})$ by mol = 75-90) are robustly constrained by the experimental data regardless of the solution models implemented.

We have added this discussion in the Phase diagram calculation section to clarify and justify our procedure in our solid solution modeling.

2) Despite this issue with the modeling, based on hard core data presented in Figure 2, it is pretty clear that at 40 GPa (it could be slightly different) the melting temperature (solidus or liquidus) is the largest for the relevant range of concentrations in the Earth. The authors then continue by relating this melting temperature with viscosity and predict a peak in viscosity at ~ 40 GPa, followed by a decrease up to ~ 50 GPa, followed again by a steady increase. If such a scaling relation between viscosity and melting temperature is valid, this is an exceedingly interesting behavior that contrasts with measurements of viscosity based deformation behavior at low temperatures (Ref. 16). This is a significant result.

Thank you.

3) The authors admit it is unclear how the spin crossover affects the viscosity. In a previously and very relevant publication (Fig. 4 in Ref. 26 in SI) another phenomenological estimate of viscosity based on the elastic strain energy model was used to speculate on the possible existence of a viscosity drop in the lower mantle above 40 GPa followed by an increase in the pressure range measured here. The cause of this behavior was discussed there as well and was related to the behavior of the activation energy for diffusion in the presence of a softening of the bulk modulus (or bond strength). It is very interesting that similar behavior for the viscosity is found now based on the melting behavior of ferropericlase, which is quite different from that put forward by Ref. 16. The similarities between the present results and that predicted in Ref. 26 of SI (Wentzcovitch *et al.*, PNAS 2008) should be presented and discussed in the main text. These results support each other in contrast to the distinct viscosity behavior proposed in Ref. 16. This discussion is very important and should be included.

Thank you for the suggestion. We have added a paragraph to discuss the consistency of the viscosity profile of ferropericlase predicted by [Wentzcovitch *et al.*, 2009] and this study in the main text. The new text is given below:

“It is noted that the viscosity profile of ferropericlase calculated in this study is in excellent agreement with that proposed by [Wentzcovitch *et al.*, 2009], where viscosity was estimated based on the elastic strain energy model. The viscosity is a function of the effective diffusivity (see equations 8, 9 in Methods) which is further related to the shear and bulk moduli by the elastic strain energy model because ionic diffusion shears and stretches the chemical bonds on the diffusion path [Wentzcovitch *et al.*, 2009]. Using the elastic constants obtained by first-principle computations, a local peak viscosity was found at ~40 GPa [Wentzcovitch *et al.*, 2009]. The consistency between [Wentzcovitch *et al.*, 2009] and this study is not fortuitous. Both studies suggest that the viscosity of ferropericlase is correlated with its thermoelastic properties. Although our observation is only directly concerned with the melting behavior of ferropericlase, melting has been suggested to be closely related to the thermoelastic properties in general by both the shear instability model and the vibrational instability model (see [Poirier, 2000] for a review). Based on the latter instability model, Lindemann’s law mentioned earlier is one example that relates the elastic constants to melting temperatures. As such, the viscosity profile of ferropericlase by ref. [Wentzcovitch *et al.*, 2009] and this study indicate that the viscosity peaking at ~40 GPa is a necessary result of the thermoelastic anomaly induced by the spin crossover of Fe²⁺ for Earth-relevant ferropericlase.”

4)Details: Line 108: “...the spin transition softens and condenses (MgFe)O”... “Densifies” is a better expression than “condenses”.

Thank you. We have changed “condenses” to “densifies”.

5)Details: Lines 111-112: “...once the spin transition to low-spin state is complete, the moduli and density will again monotonically increase at values greater than those in the high spin state...”. Actually, this is incorrect. The moduli and density start increasing monotonically after the mid-point of the spin crossover, not after it has ended.

Thanks for pointing out this confusing sentence. This sentence has been revised accordingly for clarity.

6)Details: Line 92: punctuation or parenthesis missing.

Thank you. It has now been fixed.

In summary, this paper presents good and important results on the melting behavior of ferropericlase. It is original and interesting data. As always, speculations of geophysical implications may not endure, as often is the case, but it illustrates the significance of these results to geophysics.

Thank you.

Reviewer #2 (Remarks to the Author):

General comments:

This article presents experimental measurements of the (Fe,Mg)O melting relations under high pressure and a viscosity model for the Earth’s mantle inferred from these results using homologous temperature scaling. This subject is suitable for Nature Communications. The presentation is well-written and clear. The experimental results are quite clear, even if the absence of pressure medium to determine melting properties needs more careful discussion. As a non-specialist regarding the model used here to relate melting curves and viscosity, it is difficult for me to judge the uncertainties for viscosity determination are. In conclusion, I would recommend this paper for publication with the addition of more complete discussion on the two previous points.

For the reviewer’s first point, the effects of the absence of pressure medium are discussed in detail below (2nd comment given by this Reviewer).

See the end of this response (the last comment of the 3rd Reviewer) for the second point raised in this comment.

Detailed comments:

1)The experimental determination of the Mg,FeO phase diagram and melting relations under extreme conditions is a very challenging topic. I find quite clear the discussion regarding temperature determination, and I should compliment the authors for the past effort made on the wavelength dependent absorption of the sample.

Thank you.

2)I am however quite puzzled by the absence of pressure medium in the present melting experiments. I wonder how the temperature gradients generated could contribute somehow to the error bars and then affect the presented maximum in the melting curves.

We address the reviewer's concerns as follows:

First, we want to clarify that the purpose of not using any pressure media for Mg-rich samples is to minimize the possible contamination introduced by the pressure medium. In addition, for Mg-rich samples, even if a pressure medium is used, large temperature gradients still exist due to the small absorption coefficient of the materials to laser radiation. This means that the laser can easily penetrate through the Mg-rich (Mg,Fe)O and melting would still begin from the interior of the samples, setting up temperature gradients.

For the Fe-rich samples, we use MgO as insulation and a pressure media so that we are able to heat—without which, laser heating would be difficult. We find diffusion between the MgO and Fe-rich ferropericline and so do need to take this into account when applying our temperature correction due to wavelength-dependent absorption. Fortunately, in terms of “contamination,” MgO is fine since it is already part of the binary system we are investigating. Using a noble gas such as argon, for example, may cause lowered melting temperatures due to incorporation in to melts of this composition [Du and Lee, 2014]. Additionally, due to the high temperatures anticipated for ferropericline melting, we avoided alkali halides due to strong changes in the optical properties, which cause a rapid increase in temperature near their melting points[Arveson et al., 2017; Boehler et al., 1996].

Second, the correction of the effects of temperature gradients on temperature deviation is already incorporated in our inverse modeling method (section II in [Deng et al., 2017]). To summarize the consideration in [Deng et al., 2017], we solve the steady heat flow equation with proper boundary conditions to obtain the 1D axial temperature gradient. More rigorous temperature profiles can be calculated using the TempDAC code [Rainey et al., 2013] with the knowledge of thermodynamic properties of materials at corresponding conditions. But unfortunately, those parameters are poorly constrained for most Earth materials at elevated pressures and temperatures. Nevertheless, the fine structure of the temperature profile obtained by rigorous thermodynamic simulation is not expected to change the axial temperature distribution within the melt much while it does alter the fine structure of the temperature profile of the solid part. As such, temperature correction will not be influenced largely by the rigorous temperature profile calculation since the hottest part (melt) dominates the effects of the temperature. A comparison was conducted between Fig 7 of [Rainey et al., 2013] and the solution given by the 1D thermal diffusion equation and an ~4% deviation was found, which is included in the final uncertainty reported in this study, ~10%.

Third, the effects of the uncertainties in temperatures presented as maxima in the melting curves have been discussed in the manuscript (Supplementary Fig. 4 in and Supplementary Note 4, *Interpretation of local maxima in melting temperatures* in revised manuscript). Assuming 1) the solidus melting temperatures at 40 GPa is only marginally affected by the spin transition compared with those at 50 GPa and 2) the increment of solidus melting temperatures of (Mg,Fe)O from 40 to 50 GPa without the effects of spin transition is a linear combination between the increments of melting temperatures of endmembers ($\Delta T = 70 - 280$ K for Mg# 0-100), we can discriminate the amount of temperatures depressed due to spin crossover. It is clear from Supplementary Fig. 4 in the revised manuscript that the depression in solidus melting

temperatures for (Mg,Fe)O with Mg# between 52 to 98 cannot merely be explained by the $\pm 10\%$ measurement uncertainty in the solidus melting temperature at 50 GPa.

The above arguments were briefly mentioned in the “Temperature determination” of the Method section, lines 306-316 but were discussed in detail in [Deng *et al.*, 2017]. We chose not to discuss them in further detail here due to its highly technical nature and its comprehensive treatment in our recent paper [Deng *et al.*, 2017].

3) Also, is there any quantified C contamination in the sample?

We have not tried to quantify C in our quenched samples due to limitations of the Analytical Scanning Electron Microscope (ASEM; JEOL 7600F) at the Center for Functional Nanomaterials at Brookhaven National Lab which cannot quantify the concentration of elements with $Z < 8$. However, we argue that carbon contamination > 0.01 ppm by weight in our samples is unlikely in this study and its effects on melting temperatures are negligible based on the following evidence.

The measurement totals of our quenched samples are mostly within the range of 97-103%, suggesting that C contamination, if it exists, is likely small. In addition, previous studies suggest a negligible solubility of carbon in (Mg,Fe)O. Shcheka *et al.* [2006] found that the maximum carbon solubility in most mantle silicates, including wadsleyite, ringwoodite, MgSiO_3 -ilmenite and MgSiO_3 -bridgmanite is below their limit of detection of 40–110 ppb by weight. Because (Mg,Fe)O does not have tetravalent cation, we expect that the solubility of carbon in ferropericlase should be lower than the above silicates and therefore negligible. Indeed, [Wolf and Grabke, 1985] reported that the solubility of carbon in FeO and MgO solid is under the detection limit (0.01 ppm by weight) at 1 bar. The effects of carbon on the melting of ferropericlase is unknown. But a carbon concentration < 0.01 ppm is unlikely to have any pronounced effects on the melting of (Mg,Fe)O ferropericlase.

In addition, Prakupenka *et al.* [2010] studied the high-pressure-induced carbon transport from diamond anvils and observed the carbon present in the ferropericlase (Mg,Fe)O/SiO₂ mixture as independent crystalline phases. First, we did not observe crystallized carbon phases (diamond or graphite) in our sample cross-section (see Figure 1 for representative cross sections). Second, even if carbon presents as independent minor phases, it is unknown to how it will affect the melting temperatures. A qualitative estimation is that since independent minor carbon phases do not chemically react with (Mg,Fe)O as we argued above and therefore do not contribute to the configurational entropy of the system as the dissolved carbon, the melting temperature of (Mg,Fe)O will hardly be affected. However, some other physical properties, for example, the electricity conductivity may be very sensitive to the presence of the minor carbon phases.

Furthermore, we used the same sets of starting materials and followed the same procedures for all the experimental runs, if C contamination existed and drastically affected the melting temperature, it does not make sense if the effects of C on melting temperatures only occur for runs at pressures 50, 60 and 80 GPa and not at 27 and 40 GPa.

We have included a discussion of this issue in the Chemical Analysis section of the revised manuscript, lines 311-322.

4) Also, the absence of any irregular effect on FeO and MgO melting curves in the present pressure range is difficult to correlate, in my opinion, with the present maximum in MgFeO melting, even for low Mg content.

We agree with the reviewer’s point that a local maximum in the melting curve of (Mg,Fe)O is unexpected and is difficult to correlate with the “regular” melting curve of the endmembers. But the (Mg,Fe)O does show very different physical properties compared with its two endmembers. The spin transition of Fe^{2+} is a good example. Within the pressure range (27 – 83 GPa) investigated in this study, Fe^{2+} of pure FeO do not show any spin transition whereas $(\text{Mg}_x\text{Fe}_{1-x})\text{O}$ with $x > 0.6$ does [Glazyrin *et al.*, 2016]. As argued in the manuscript, Lindemann’s law relates the melting temperature to the thermo-elastic properties, which has

been shown to be sensitive to the spin crossover. Therefore, the fact that no experimental results show the depressed melting curves of FeO in this pressure range is not surprising [Fischer and Campbell, 2010; Knittle and Jeanloz, 1991]. In other words, the melting behavior of $(\text{Mg}_x\text{Fe}_{1-x})\text{O}$ can not be simply treated as the average of those of pure FeO and MgO owing to the different response of Fe^{2+} to pressures. Indeed we show that taking a linear interpolation of the melting temperatures of FeO and MgO yields very different values for the solidus of ferropericlasite (>1000 K, Supplementary Fig. 3 in the revised manuscript).

Reviewer #3 (Remarks to the Author):

Deng and Li, 2017 Nature Communications

reject

The authors carried out high-pressure melting experiments on (Mg,Fe)O ferropericlasite (and magnesiowüstite) up to 83 GPa and constrained melting phase relations of MgO-FeO system. The obtained solidus and liquidus curves exhibit local maxima at about 40 GPa, which may cause viscosity jump in the lower mantle.

Since (Mg,Fe)O is second most abundant mineral in the Earth's lower mantle, its melting temperature could affect the melting temperature and viscosity of the lower mantle. I agree that the wavelength-dependent absorption must be considered in order to determine temperature by spectroradiometric method. Thermodynamic treatment to get phase relation of (Mg,Fe)O seems valid. However, since viscosity is not only sensitive to the melting temperature of consisting minerals, but also many other factors as the authors stated in the manuscript, the local maxima in melting curve of ferropericlasite has been just a possible cause of the jump in viscosity. The present data is new, but I find no new scientific ideas in this manuscript. This manuscript should be submitted to professional journals such EPSL, PEPI and JGR rather than Nature Communications.

We agree that viscosity is sensitive to many factors and are aware that there are many other mechanisms that have been proposed to account for the viscosity jump in the shallow lower mantle [e.g., Ballmer et al., *Nature Communications*, 2017; Shim et al., *PNAS*, 2017]. We also agree that the present data are new and unexpected, and thus necessarily provoke new ideas on how this may influence viscosity in the mantle. This is where our manuscript comes into play.

In addition, I have some minor comments on this manuscript.

1. Fine-grained (Mg,Fe)O samples were used in this study. Did the authors take care with contamination of moisture in the samples? Water in the sample drastically decreases its melting temperature as we know. Also, I wonder if the authors checked carbon contamination during laser heating experiments by using in-situ XRD or ex-situ chemical analyses.

In order to minimize moisture contamination, the starting materials were kept in a vacuum oven (80 °C) except being taken out for sample loading. The sample assembly was also oven-dried after cell loading and prior to pressurization.

As mentioned above, we have not performed ex-situ chemical analyses to check for C. However, given our good total values (97-103%), we don't expect to have a lot of C contamination, although it cannot be ruled out. We also have not checked for C contamination using in-situ XRD. See the response above for more details (the 3rd comment given by the 2nd reviewer).

We have included additional text in the Methods section (lines 294-297 and 311-322) that addresses both forms of potential contamination.

2. Interpretation of the observed unusual melting behavior at ~40 GPa may be the spin crossover in (Mg,Fe)O. Spin transition zone spreads with increasing temperature as many experimental and theoretical studies reported (e.g., Lin et al., 2007 Science; Tsuchiya et al., 2004 PRL). However, the authors referred to the spin transition pressure range of 300 K as shown in Figs. 2 and 3. The authors discuss pressure response of melting curves, and should refer the spin transition zone at high temperatures.

We agree with the reviewer that the spin transition pressure range at high temperatures is more relevant in the interpretation of the observed anomalous melting behaviors of ferropericlase. However, we also argue that the spin transition pressure range at 300 K is a good approximation and is appropriate to use it here based on the following points.

1) The effects of temperature on the spin transition pressure range for ferropericlase are not yet fully understood. First, the spin transition onset pressures have been repeatedly found to be temperature independent or nearly independent by both experiments [*Lin et al.*, 2007; *Lyubutin et al.*, 2013; *Müller et al.*, 2017] and simulations [*Ghosh and Karki*, 2016; *Tsuchiya et al.*, 2006; *Wu et al.*, 2009] although the simulation by [*Holmström and Stixrude*, 2015] suggests a strong delay of onset pressure with increasing temperatures. Second, the preceding studies all agree that the spin transition completion pressures increase with temperatures. However, the specific value of the completion pressure is highly controversial and no consensus has been found. Additionally, the composition coverage in these studies is poor. Therefore, it is almost impossible for us to determine definite and consistent spin transition completion pressures at high temperatures for ferropericlase of the four different compositions (Mg# 75, 80, 85, 90) presented in Figs. 3 and 4 (Figs. 2 and 3 in the old manuscript).

2) The spin transition onset and completion pressures of ferropericlase at ambient temperature have been extensively studied for various compositions. The results have been compiled by [*Glazyrin et al.*, 2016] and a clear trend of pressure vs. composition has been identified. As argued above, the onset pressures at 300 K are presumably very close to those at high temperatures while the completion pressures at high temperatures are possibly higher. Nevertheless, the comparison between the spin transition pressures range and melting curves is to qualitatively illustrate that the pressures corresponding to the melting temperature depression overlaps with the spin transition pressures. The broader spin transition pressure range that starts at similar pressures as at 300 K should cover the pressure range we presented in the Figs. 3 and 4 (Figs. 2 and 3 in the old manuscript).

To sum up, using the spin transition pressure range at high temperatures will not contribute to the purpose of the comparison but introduce more uncertainties due to the poorly-constrained spin transition completion pressure.

The following sentence has now been added in the captions of Figs. 3, 4 and Supplementary Fig. 5 in the current manuscript to illustrate the above points:

“The spin transition pressure ranges at corresponding high temperatures are likely broader but begin at pressures similar to those at 300 K [*Lin et al.*, 2007; *Tsuchiya et al.*, 2006].”

3. As discussed in Ohta et al. (2017 EPSL 465, 29-37), within such a spin transition zone of ferropericlase, ferropericlase can be regarded as three component mixtures of MgO, high-spin FeO, and low-spin FeO, which induce the reduction of melting temperature of ferropericlase.

Thanks for bringing this interesting work to our attention. The idea that ferropericlase can be considered as a ternary mixture of MgO, high-spin FeO and low-spin FeO has also been proposed by [*Speziale et al.*, 2005]. But the statement that a ternary mixing of MgO, high-spin FeO and low-spin FeO will induce the reduction of melting temperature of ferropericlase is not as intuitive. As far as we can tell, this statement was also neither mentioned in [*Speziale et al.*, 2005] or [*Ohta et al.*, 2017]. Nevertheless, qualitatively, one more component induces additional configurational entropy, which may depress the melting temperature similar as the addition of impurities. Once the spin transition is complete, however, we are back to a binary with smaller configurational entropy and higher temperatures. In any case, the spin transition is the fundamental cause of the melting curve depression.

4. Holmstrom and Stixrude (2015 PRL) suggested metallization of ferropericlase with 25 mol% Fe at high pressures and temperatures. Is the metallization of ferropericlase another cause of anomaly in melting curves that the authors observe?

We consider the metallization of ferropericlase with 25% Fe reported by [Holmström and Stixrude, 2015] as unlikely to account for the anomaly in melting curves observed in this study for the following reasons.

First, the electrical conductivity of ferropericlase at core-mantle boundary conditions predicted by [Holmström and Stixrude, 2015] is two orders of magnitude higher than the later experimental observations by [Ohta *et al.*, 2017]. The latter suggests that the effects of pressure on the electrical conductivity of ferropericlase diminish with increasing temperatures and become negligibly small at the ~2000 K (see their Fig. 6). As such, at the temperatures reached in this study (>3000 K), the controlling factors for electrical conductivity is temperature instead of pressure. In contrast, our measurements show that the melting curve anomalies are sensitive to the experimental pressures. The different dominant factors may indicate different mechanisms for metallization and melting curve depression.

However, we do not exclude the possibility that metallization of ferropericlase is correlated with the melting curves anomalies observed. Indeed, the electrical conductivity vs. pressure plots show strong depression within the pressure range where the spin transition occurs for ferropericlase of various compositions [Ohta *et al.*, 2007; Ohta *et al.*, 2014], resembling the melting curves observed in this study. In fact, both the electrical conductivity depression and, now, the melting temperature depression of ferropericlase have been ascribed to the spin crossover of Fe^{2+} .

To sum up, the metallization of ferropericlase with 25% Fe reported by [Holmström and Stixrude, 2015] contradicts with the experimental results [Ohta *et al.*, 2017] and is unlikely to account for the melting temperature anomalies, while the depression of both the electrical conductivities and melting temperatures for ferropericlase could be due to the spin crossover.

5. There is one new paper regarding experimentally-determined melting curve of MgO. Kimura et al., 2017 Nature Communications, Melting temperatures of MgO under high pressure by micro-texture analysis. I ask the authors to cite it.

Thank you. We have now included [Kimura *et al.*, 2017] in the revised manuscript.

6. Error bars of temperature should be added in Figs. 2 and 3. In Figs. 2 and 3, spin transition range of high temperature should be shown rather than that at 300 K.

In the original version of the manuscript we chose not to show the error bars in Figs. 3 and 4 of the current manuscript (Figs. 2 and 3 in the original manuscript) for clarity, but instead we included in the caption of in Figs. 3 (Figs. 2 in the old manuscript), “Uncertainties on the temperatures are $\sim\pm 10\%$.” We have now included the uncertainties for Fig. 3 (the shaded grey region) in the figure below. We ask the editor which version should be included in the manuscript should it be accepted.

For Fig. 4 (Figs. 3 in the old manuscript), the wide range of compositions, the two different geotherms and the two different solid solution models show the spread of viscosity values versus depth, which implicitly shows the range of uncertainty in viscosity. This consideration has been added to the caption to the Fig. 4 and Supplementary Fig. 5 in the revised manuscript.

Reference:

- Arveson, S., B. Kiefer, Z. Liu, and K. K. M. Lee (2017), Thermally induced coloration of KBr at high pressures, *Under review*.
- Boehler, R., M. Ross, and D. B. Boercker (1996), High-pressure melting curves of alkali halides, *Phys Rev B*, 53(2), 556-563.
- Coppari, F., R. F. Smith, J. H. Eggert, J. Wang, J. R. Rygg, A. Lazicki, J. A. Hawreliak, G. W. Collins, and T. S. Duffy (2013), Experimental evidence for a phase transition in magnesium oxide at exoplanet pressures, *Nat Geosci*, 6(11), 926-929.
- Deng, J., Z. Du, L. R. Benedetti, and K. K. M. Lee (2017), The influence of wavelength-dependent absorption and temperature gradients on temperature determination in laser-heated diamond-anvil cells, *J Appl Phys*, 121(2).
- Du, Z., and K. K. M. Lee (2014), High-pressure melting of MgO from (Mg,Fe)O solid solutions, *Geophysical Research Letters*, 41(22), 8061-8066.
- Fischer, R. A., and A. J. Campbell (2010), High-pressure melting of wüstite, *American Mineralogist*, 95(10), 1473-1477.
- Fischer, R. A., A. J. Campbell, O. T. Lord, G. A. Shofner, P. Dera, and V. B. Prakapenka (2011), Phase transition and metallization of FeO at high pressures and temperatures, *Geophysical Research Letters*, 38(24), L24301.
- Ghosh, D. B., and B. B. Karki (2016), Solid-liquid density and spin crossovers in (Mg, Fe)O system at deep mantle conditions, *Sci Rep-Uk*, 6, 37269.
- Glazyrin, K., N. Miyajima, J. S. Smith, and K. K. M. Lee (2016), Compression of a multiphase mantle assemblage: Effects of undesirable stress and stress annealing on the iron spin state crossover in ferroperricite, *Journal of Geophysical Research: Solid Earth*, 121(5), 2015JB012321.
- Holmström, E., and L. Stixrude (2015), Spin Crossover in Ferroperricite from First-Principles Molecular Dynamics, *Phys Rev Lett*, 114(11), 117202.
- Kimura, T., H. Ohfuji, M. Nishi, and T. Irifune (2017), Melting temperatures of MgO under high pressure by micro-texture analysis, *Nature Communications*, 8, 15735.
- Knittle, E., and R. Jeanloz (1991), The High-Pressure Phase-Diagram of Fe_{0.94}O - a Possible Constituent of the Earth's Core, *Journal of Geophysical Research-Solid Earth*, 96(B10), 16169-16180.
- Lin, J.-F., G. Vankó, S. D. Jacobsen, V. Iota, V. V. Struzhkin, V. B. Prakapenka, A. Kuznetsov, and C.-S. Yoo (2007), Spin Transition Zone in Earth's Lower Mantle, *Science*, 317(5845), 1740-1743.
- Lyubutin, I. S., et al. (2013), Quantum critical point and spin fluctuations in lower-mantle ferroperricite, *Proceedings of the National Academy of Sciences of the United States of America*, 110(18), 7142-7147.
- Müller, J., I. Efthimiopoulos, S. Jahn, and M. Koch-Müller (2017), Effect of temperature on the pressure-induced spin transition in siderite and iron-bearing magnesite: a Raman spectroscopy study, *Eur J Mineral.*
- Ohta, K., K. Hirose, S. Onoda, and K. Shimizu (2007), The effect of iron spin transition on electrical conductivity of (Mg,Fe)O magnesiowüstite, *Proceedings of the Japan Academy. Series B, Physical and Biological Sciences*, 83(3), 97-100.
- Ohta, K., T. Yagi, K. Hirose, and Y. Ohishi (2017), Thermal conductivity of ferroperricite in the Earth's lower mantle, *Earth and Planetary Science Letters*, 465, 29-37.
- Ohta, K., K. Fujino, Y. Kuwayama, T. Kondo, K. Shimizu, and Y. Ohishi (2014), Highly conductive iron-rich (Mg,Fe)O magnesiowüstite and its stability in the Earth's lower mantle, *Journal of Geophysical Research: Solid Earth*, 119(6), 2014JB010972.
- Poirier, J. P. (2000), *Introduction To The Physics Of The Earth's Interior*, 2nd ed., Cambridge University Press, New York.
- Prakapenka, V., G. Y. Shen, and L. Dubrovinsky (2010), *Carbon transport in diamond anvil cell*, 237-249 pp.

Rainey, E. S. G., J. W. Hernlund, and A. Kavner (2013), Temperature distributions in the laser-heated diamond anvil cell from 3-D numerical modeling, *J Appl Phys*, 114(20), 204905.

Shcheka, S. S., M. Wiedenbeck, D. J. Frost, and H. Keppler (2006), Carbon solubility in mantle minerals, *Earth and Planetary Science Letters*, 245(3-4), 730-742.

Speziale, S., A. Milner, V. E. Lee, S. M. Clark, M. P. Pasternak, and R. Jeanloz (2005), Iron spin transition in Earth's mantle, *Proceedings of the National Academy of Sciences of the United States of America*, 102(50), 17918-17922.

Tsuchiya, T., R. M. Wentzcovitch, C. R. S. da Silva, and S. de Gironcoli (2006), Spin transition in magnesiowustite in earth's lower mantle, *Phys Rev Lett*, 96(19).

Wentzcovitch, R. M., J. F. Justo, Z. Wu, C. R. S. da Silva, D. A. Yuen, and D. Kohlstedt (2009), Anomalous compressibility of ferropericlase throughout the iron spin cross-over, *Proceedings of the National Academy of Sciences*, 106(21), 8447-8452.

Wolf, I., and H. J. Grabke (1985), A study on the solubility and distribution of carbon in oxides, *Solid State Commun*, 54(1), 5-10.

Wu, Z., J. F. Justo, C. R. S. da Silva, S. de Gironcoli, and R. M. Wentzcovitch (2009), Anomalous thermodynamic properties in ferropericlase throughout its spin crossover (vol 80, 014409, 2009), *Phys Rev B*, 80(9).

REVIEWERS' COMMENTS:

Reviewer #2 (Remarks to the Author):

The effort made by the authors to answer to the different queries made by the 3 reviewers is really excellent. I would just the authors to insert the comments made in the answers to the reviewers regarding the absence of pressure medium in the Method section or in the supplementary materials.

I would therefore accept the paper for publication.

REVIEWERS' COMMENTS:

Reviewer #2 (Remarks to the Author):

The effort made by the authors to answer to the different queries made by the 3 reviewers is really excellent. I would just the authors to insert the comments made in the answers to the reviewers ragrding the absence of pressure medium in the Method section or in the supplementary materials. I would therefore accept the paper for publication.

Thank you. We have incorporated our last response to the comment regarding the absence of pressure medium as Supplementary Note 6.